


**Sinkholes of the Dead Sea Basin: A Result of Anthropogenic**
**Disturbance to Nature and Sign for More and Greater Hazards**
Hilmi S. Salem
Sustainable Development Research Institute
Bethlehem, West Bank, Palestine (Occupied)
*Correspondence to*: Hilmi S. Salem (hilmisalem@yahoo.com)
**Abstract** Over the last few decades, thousands of sinkholes have developed at an increasing pace along the western
and eastern shores of the Dead Sea. Recent studies indicate that the number of sinkholes in the Dead Sea Basin
(DSB) has reached more than 6,000, with 1–10 m deep and up to 25–30 m in diameter, on average, each. These
sinkholes can open-up suddenly and swallow whatever found above them, resulting in an area that looks like an
earthquake zone. Sinkholes in the DSB are formed when a subterranean salt layer that once bordered the Dead Sea is
dissolved by underground freshwater that follows the migration of the saltwater–freshwater interface, due to
receding water level of the Dead Sea. Consequently, large areas of land are subsiding, causing the formation of
sinkholes in the region. Also, based on the fact that the Dead Sea's region is tectonically and seismically active, as it
is greatly affected by the Dead Sea Transform Fault System in the region, sinkholes can also be evolved as a result
of tectonic and seismic activities. Nevertheless, sinkholes, as geomorphologic features occurring in the DSB,
represent the most remarkable evidence of the brutal interferences of humans in the Dead Sea, especially sinkholes
started to appear remarkably and frequently in the last half a century only. They present a serious problem to the
Dead Sea's region, as they have led to damages in the infrastructures, and have threatened the safety of humans.
Large coastal areas along the Dead Sea shores on both sides of the Sea have already been totally closed to access,
because occurrence of more sinkholes is continually developing and rapidly increasing in number. This paper
investigates and discusses the occurrence of sinkholes in the Dead Sea Basin, geomorphologically, geologically,
seismically, limnologically, and socioeconomically, as well as with respect to the steady decline, at alarming rate, of
the Dead Sea's water level and the continuous shrinkage of its surface area, as the Dead Sea's water level has been
declining, on average, one meter per year just during the last 50 years or so. Also, the sinkholes' occurrence in the
DSB is investigated and discussed, with respect to tectonics and seismicity affecting the region. The results indicate
that more attention should be paid to this phenomenon – the rapid and intensive occurrence of sinkholes in the Dead
Sea's region – since it is a serious and dangerous disaster, affecting the Dead Sea Basin itself and its surrounding
environment. Unfortunately, this disaster is predominantly caused anthropogenically, as a result of the decline of the
Dead Sea's water level caused by humans' activities. So, this problem can be avoided or possibly reduced if man
would think about nature's protection rather than about his own interests.
**Keywords:** *Sinkholes; Dead Sea Basin; Dead Seas' Water Level's Decline and Surface Area's Shrinkage;*
*Anthropogenic Causes; Brine and Freshwater; Tectonics and Seismicity (Earthquakes); Dead Sea Transform Fault*
*System*



## 1. Introduction

A sinkhole, also known as sink, stream-sink, cenote, swallet, swallow hole, ponor, or doline, is defined as a geomorphological depression or hole in the ground, caused by some forms of collapse of the surface layers because of different reasons (Goudie 2006). The morphology of sinkholes and their genetic mechanisms, spatial distribution, and associated risks are well known (Gunay et al. 2000; Soriano and Simo´n 2002; Galve et al. 2009; Wonder Mondo 2010; Soriano et al. 2012; Galve et al. 2015; Gutiérrez 2016; Aden 2018).

Sinkholes can form when the land surface topography is changed. Karst is a topography formed from the dissolution of soluble rocks, such as carbonates (limestone and dolomite) and salts (halite, gypsum, and anhydrite). Ground collapses, subsidence, and contamination are hazards strongly associated to water, often encountered in a karst environment. Most of the sinkholes are caused by karst processes, including the 'karstification process,' meaning that rocks are chemically dissolved as happened in carbonate rocks (Lard et al. 1995; Dubois et al. 2014); the 'suffosion process,' meaning that rocks are subsiding and collapsing as happened in sandstone rocks (Waltham and Lu 2007; Parise 2008; BGS 2017); and 'dissolution process' as happened in salt rocks, similar to the case of the Dead Sea's salt layers (Frumkin et al. 2001; Shalev et al. 2006; Salem 2009; Frumkin et al. 2011; Watson et al. 2018). Karstification, suffosion, and dissolution processes require soluble rocks; favorable climatic conditions; structural elements that act as conduits (such as faults, ruptures, fractures, flexures, fissures, etc.); and a hydraulic gradient; as well as high porosity and permeability (or hydraulic conductivity), which all facilitate water mobility. Therefore, a good knowledge of the distribution of groundwater potentiometric heads, and fluxes in space and time, as well as the other mentioned-factors is necessary to predict and probably prevent karst hazards. Aside from the fact that sinkholes are naturally caused, humans are also responsible for the formation of sinkholes. Activities like drilling, mining, construction, broken water or drain pipes, improperly compacted soil after excavation work or even heavy traffic can result in small to large sinkholes. Water from broken pipes can penetrate through mud and rocks and erode the ground underneath and cause sinkholes. Sometimes, heavy weight on soft soil can result in collapse of ground, resulting in a sinkhole. The sinkholes resulted from these natural and anthropogenic processes vary, generally, in size from 1–600 m (3.3–2,000 ft) both in diameter and depth, and some of them are even larger than that, as indicated below. They also vary in shape from soil-lined bowls to bedrock-edged chasms.

Sinkholes may form gradually or suddenly, and are found worldwide. They are found, for instance, in Europe (Czech Republic, Croatia, Serbia, Italy, Greece, England, and Ireland); in Africa (Egypt, South Africa, and Zambia); in the Caribbean (Bahamas); in North America (Mexico, USA, and Canada); in Central America (Guatemala and Belize); in South America (Venezuela); in Oceana (New Zealand and Papua New Guinea); and in Asia (China, Turkey, Lebanon, and Oman). In Oman, for example, there are the Wadi Shab and Bimmah sinkhole, the Dibab sinkhole, and the Teiq (Taiq, Teeq, Tayq) sinkhole. The Teiq Sinkhole, in Salalah – Dhofar, Oman, is one of the largest sinkholes in the world, where several perennial wadis (valleys) fall with spectacular water-falls into this limestone sinkhole, which is approximately 210-m (≈ 670 ft) deep and 130–150 m in diameter (Muscat Daily 2015).



It is worth-mentioning that the sinkholes in Oman, despite of their importance to tourism, they are not studied, and,
thus, there is no single scientific article published in refereed journals about them. The largest sinkhole, that has
been reported so far, is the 'Xiaozhai Tiankeng Sinkhole' in China, which is up to 662 m deep (≈ 2,172 ft), with
nearly vertical walls. This sinkhole is one of the most impressive natural attractions on Earth, which had been carved
out in limestone rocks by a powerful underground river (Xuewen and Weihai 2006; Wonder Mondo 2010; Starr
2018). A Chinese-British expedition's team that surveyed this giant sinkhole can be seen on this short YouTube
(1.31 minutes' long): https://youtu.be/j4hjkZfEdUU?t=4.
Remarkably, there are also under-sea (underwater) sinkholes, also known as 'blue holes' (Osborne 2017; G&Y
2018). These blue holes are large marine sinkholes that mostly formed during past ice ages when sea levels were
much lower than that in present time. They were subject to the same process of erosion from rain and chemical
weathering as any other area that have sinkholes. After being submerged, the erosion ceased, and the deep blue
caverns (sinkholes) are left. Some examples on the underwater sinkholes (blue holes) are studied by Palozzi et al.
(2010) in Italy, Biddanda et al. (2011) in the USA (Lake Huron – one of the Great Lakes), and Medina-Moreno et al.
(2014) in Mexico, as well as reported by Tennenhouse (2016) in China, and by Shepert (2018) in Canada. There are
no scientific publications available on the last two examples of blue holes in China and Canada.
According to La Rosa et al. (2018), more than 40% of the sinkholes of Italy are found in seismically hazardous
zones. However, according to La Rosa et al. (2018), it remains unclear whether seismicity in that region may trigger
sinkholes' collapse or not. La Rosa et al. (2018) used a multidisciplinary data set of Interferometric Synthetic
Aperture Radar (InSAR), surface mapping, and historical records of sinkholes' activity to show that the Prà di Lama
Lake is a long-lived sinkhole that was formed in an active fault zone and grew through several events of unrest,
characterized by episodic subsidence and lake-level changes. Moreover, InSAR measurements showed that
continuous subsidence at rates of up to 7.1 mm/yr occurred during 2003–2008, between events of unrest. However,
earthquakes on the major faults near the sinkhole do not trigger sinkhole activity, but low-magnitude earthquakes at
4–12 km depth occurred during sinkhole unrest in 1996 and 2016 (La Rosa et al. 2018). These observations were
interpreted as evidence of seismic creep at depth, causing fracturing, and ultimately leading to the formation and
growth of the Prà di Lama sinkhole (Lake) (La Rosa et al. 2018). In Japan, a massive sinkhole has appeared in a
Japanese road after an earthquake centered on Osaka, which caused substantial destruction to regional infrastructure,
as well as at least four deaths (Mcgrath 2018). This sinkhole resulted in hundreds of people being injured, walls
being knocked over, and fires triggered in Japan's second-most populous city – Osaka. In addition, 170,000 homes
were left without power, and flights in and out of the city's airport were grounded. That was a result of the 6.1-
magnitude earthquake that hit Osaka city, Japan, on 18 June 2018 (Mcgrath 2018). So, sinkholes can be directly or
indirectly triggered and/or caused by seismic events, including micro- and macro-scaled earthquakes.
Researchers have currently applied advanced techniques to investigate sinkholes. For instance, Al-Kouri et al.
(2013) used Geiographic Information System (GIS) and Remote Sensing's techniques, including a Spatial Multi-



Criteria Evaluation's (SMCE) approach to produce a geo-hazard's map for the limestone sinkholes in the Kinta
Valley, northeastern Malaysia. Those sinkholes were resulted, over the last 3 decades, anthropogically because of
the uncontrolled land-use and development activities that have led to significant changes in topography and
geomorphology, causing the appearance of the sinkholes. Sparavigna (2016) used Satellite Image Time Series
(SITS) with high resolution to investigate the decreasing level of the Dead Sea water and the occurrence of
sinkholes in the region. Sparavigna (2016) concluded that the SITS clearly showed that the number of sinkholes is
rapidly increasing, making the shores of the Dead Sea a more dangerous place today than in the past, and it is drying
up. Goldshleger et al. (2017) developed methods for prediction of sinkholes' appearance by using mapping and
monitoring methods, based on active and passive remote-sensing means. These methods are based on measurements
from several instruments, including field spectrometry and geophysical instruments, including Ground-Penetrating
Radar (GPR) and Frequency Domain Electro-Magnetic (FDEM). These measurements were undertaken at different
time points to monitor the progress of an 'embryonic' sinkhole (a progressing one), and found that higher electrical
conductivity and soil moisture characterize the site of that progressing sinkhole. Benito-Calvo et al. (2018) explored,
for the first time, the application of a Terrestrial Laser Scanner (TLS) and a comparison of point clouds in the 4D-
monitoring of active sinkholes. Their approach was tested in three highly-active sinkholes in northeast Spain, related
to the dissolution of salt-bearing evaporites overlain by unconsolidated alluvium. The sinkholes are located in
urbanized areas and have caused severe damage to critical infrastructure (flood-control dike, a major highway).
The Dead Sea Basin (DSB) is a hub for thousands of sinkholes, representing a remarkable phenomenon that has
been developed in the last five decades. The rapid development of such a phenomenon around the Dead Sea in the
last few decades poses a major geological and environmental hazard to the local population, agriculture, and
industry (e.g. El-Isa et al. 1995; Arkin and Gilat 2000; Salameh and El-Naser 2000a; Salameh and El-Naser 2000b;
Taqieddin et al. 2000; Abou Karaki et al. 2005; Closson et al. 2007; Closson and Abou Karaki 2014; Al-Halbouni et
al. 2017; Al-Halbouni et al. 2018; Ezersky et al. 2017; Fiaschi et al. 2017; Abou Karaki et al. 2018; Polom et al.
2018). Recently, some studies were conducted on the Dead Sea Basin (DSB) to investigate the sinkholes in the
region. Al-Halbouni et al. (2018), for instance, conducted a study that included a first low altitude (< 150 m above
the ground) aerial photogrammetric survey with a Helikite Balloon at the sinkhole area of Ghor Al-Haditha, on the
eastern shore of the Dead Sea, Jordan. Al-Halbouni et al.'s (2018) study revealed a km-scale sinuous depression
bound partly by flexures and faults. The estimated minimum volume loss of this subsided zone is 1.83 million cubic
meter (MCM) with an average subsidence rate of 0.21 m/yr over the last 25 years. In the same study, Al-Halbouni et
al. (2018) conducted a numerical simulation model of the interaction of cavity growth, host material deformation,
and overburden collapse, in order to have a better understanding of sinkhole's hazards. They compared the results of
their model with the Ghor Al-Haditha sinkhole's site on the ground, and found that the observed distribution of
sinkholes' depth/diameter values in each material type (mud, alluvium, and salt) may partly reflect sinkholes'
growth trends. Also, systematic high temporal and spatial resolution's InSAR observations, incorporated with and
refined by detailed Light Detection and Ranging (LiDAR) measurements of sinkholes in the DSB, were conducted,
in order to resolve temporal and spatial relationships between gradual subsidence and sinkholes' collapse in the DSB





(Atzori et al. 2015; Nof et al. 2019). That was done to detect minute precursory subsidence before the catastrophic collapse of the sinkholes, and also to map zones susceptible to future sinkholes' formation. Currently, InSAR–LiDAR-derived subsidence maps are fundamentally used for sinkhole early warning and mitigation along the western shore of the Dead Sea, which are incorporated in all sinkholes' potential maps that are mandatory for the planning and licensing of new infrastructure in the region.

**2. Study Purpose and Methodology**

This paper deals with one of the most serious and dramatic occurrences of the Earth's surface, in one of the most amazing places on the Earth: the sinkholes of the Dead Sea. The paper's main target is to describe this hazardous phenomenon and its several social and economic implications. In this case, it is believed that the sinkholes in the Dead Sea region have been steadily evolving as a result of anthropogenic (man-made) acts, based on the facts and arguments presented and discussed herein.

This paper investigates the sinkholes in the Dead Sea Basin (DSB), and presents the resulting evaluations to all of those who are concerned with the issue of sinkholes and geo-hazards at academic, research and developmental, industrial, governmental, and nongovernmental institutions.

To achieve the goals of this paper, a wide range of up-to-date scientific and technical studies, dealing with several areas of expertise, have been comprehensively reviewed, analyzed, and cited in this paper. Field visits were also carried out, and available data was analytically used.

**3. Data and Observations**

**3.1 Geological–Geophysical Setting (Tectonics and Seismology) of the Dead Sea Basin**

The Dead Sea Basin (DSB) is part of a seismically active region that locates between two mobile tectonic plates: 1) The African Plate (including the Sinai Peninsula Sub-Plate) to the south and southwest of the Dead Sea; and 2) The Arabian Plate to the north and northeast of the Dead Sea (Figure 1). The existence of the DSB between these two mobile active plates and between two major active faults, bordering the DSB from the west and the east, has made the DSB a very active region, tectonically and seismically.



**Figure 1: Left: The tectonic framework of the Dead Sea Basin (DSB) and adjacent areas; *Right:* Digital shaded-relief map (after Darkal et al. 1990; Lunina et al. 2005).**

As shown in Figure 1 and Figure 2, the Dead Sea is bordered by two major strike-slip faults on the west and the east. The fault on the west is known as the 'Jericho Fault,' located in Historical Palestine; and the fault on the east is known as the 'Wadi Araba Fault,' (also known as Araba (or Arava) Fault), located in Jordan. The common model of the DSB, which describes its structure, is of a 'Pull-Apart Basin,' (Garfunkel and Ben-Avraham 1996) affected by both fault systems (the Jericho Fault and the Araba Fault) on both sides of the DSB. The DSB is a long ($\approx$ 150 km), narrow ($\leq$ 15 km), and deep ($<$ 8.5 km) basin, located along the Dead Sea's Transform Fault (DSTF) (Brink and Flores 2012; Wetzler et al. 2015).



The Dead Sea Basin is divided into two main sub-basins, which are the northern sub-basin and the southern sub-
basin. The northern one is larger and deeper than the southern one, whereby both sub-basins are separated by the
Lisan Peninsula. The Lisan Peninsula is underlain by a large salt diaper, with a thickness of about 8 km, a length of
up to 20 km, and a width of about 7 km, extending under the Dead Sea (Bender 1968; Al-Zoubi and Brink 2001; Al-
Zoubi and Ben-Avraham 2002; Choi et al. 2011), and, according to recent studies, it shows evidence of
hydrocarbons (Coleman and Brink 2016). The Dead Sea itself is located within a tectonic rift (known as the 'Jordan
Rift Valley' or 'Jordan Valley' – JRV), forming a topographic depression (known as 'Graben') with a width of 15–
25 km, extending from the Gulf of Aqaba on the Red Sea in the south, to Lake Tiberias (Sea of Galilee), in the
north. The JRV, formed in the Miocene Epoch (23.8–5.3 million years ago), is mainly covered by playa deposits
(salt, sand, and mud) and sand dunes. The highlands of the JRV consist of much older rocks to young deposits,
ranging in age from Precambrian in the south to Tertiary in the north.

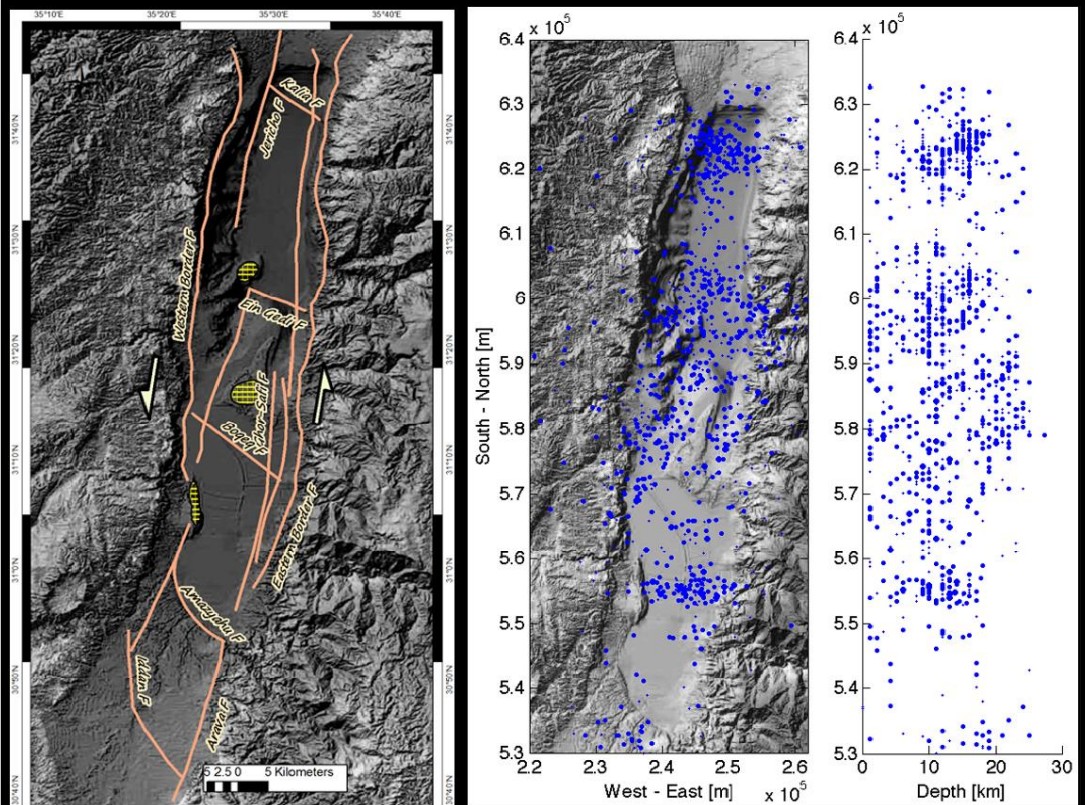



**Figure 2: Left: Map of the Jericho Fault and Araba Fault on both sides of the Dead Sea Basin (DSB), as well as some**
**other smaller faults in the DSB; Middle: A map view of the seismicity (earthquakes; blue dots) along the DSB; Right: A**
**depth north-south cross-section of the Dead Sea's seismicity (after Wetzler et al. 2012).**
Hofstetter et al. (2012) obtained a velocity structure of the crust across the DSB, by applying a tomography-based
method to local earthquakes. They used compressional wave (P-wave) travel-time of 614 earthquakes that occurred
in the DSB during a period of 26 years (1983–2009). As a result, Hofstetter et al. (2012) found that, at all depths, the
DSB is characterized by lower velocities relative to both the eastern and western sides of the DSB. They also found
that a significant seismic activity is taking place at a depth of about 20 km, mainly in the center of the DSB and its
northern part. At shallower depths (i.e. < 15 km), there is more seismic activity on the eastern side of the DSB than
on the western side. They also found that the northern part of the DSB (or Northern DSB) is generally more active
than its southern part (or Southern DSB). Asymmetry is also observed in the fault systems that border the DSB from
the west and the east (the Jericho Fault and the Araba Fault). The Araba Fault System on the eastern side of the DSB
appears to be a clear boundary at all depths down to about 20 km. Meanwhile, the depth extension of the Jericho
Fault System, on the western side of the DSB, is definitely limited to less than 15 km.
The concentration of earthquakes in the central part of the DSB (Figure 2; Middle) at depths larger than 15 km
suggests that both of the western and eastern faults of the Dead Sea act, at those depths, as one single fault that is
located in, or near, the central axis of the DSB, which is widely known as the 'Dead Sea Transform Fault' (DSTF),
also known as 'Dead Sea Transform' (DST) or Dead Sea Fault System (DSFS). The DSTF is a major fracture zone
and physiographic feature that extends from the northern Red Sea to the Taurus Mountains in Turkey, along a
distance of some 1,000 km. It is a strike-slip fault system, which currently accommodates 5–7 mm/yr of left-lateral
motion, over the past 5 million years, between the Arabian and African tectonic plates (DESERT Group et al. 2004;
Ben-Avraham et al. 2008; Le Beon et al. 2008). A total offset of 105–110 km has accumulated along the plate
boundary since the Middle Miocene (17–18 million years) (Quennell 1958; Freund et al. 1970; DESERT Group et
al. 2004). The Dead Sea Basin has been accumulating sediments since the formation of the plate boundary (Calvo
and Bartov 2001; Marco 2007) and continues to subside at present as evidenced by its low surface elevation, which
is currently at approximately 430 m below mean sea level (MSL), as discussed below.
Studying this fault system (DSFS) is of fundamental significance to the Earth Sciences (USGS 2018). Continental
transform faults, such as the DSTF (DSFS), provide a simple setting, in which deformation as a function of rock
properties, temperature, and pressure of the continental crust, can be studied. These studies are important to the
understanding of the long-term strength of the continental lithosphere, subsidence of sedimentary basins, and the
earthquakes' deformation cycle, as well as the sinkholes (developed and developing) in the Dead Sea Basin (DSB).
**3.2 Unique Characterizations and Biodiversity of Dead Sea Basin**





The Dead Sea receives its water from several sources, mainly the Jordan River, which is more than 360 km long,
and from its tributaries, including Yarmouk, Hasbani, Dan, and Banias, as well as from wadis, including Zerqa,
Mujib, Kerak, and Hasa. The Dead Sea has a catchment area of 41,650 km$^2$, current surface area of 605 km$^2$,
maximum depth of 378 m, average depth of 147 m, maximum length of 76 km, maximum width of 18 km, water
density of 1.24 kg/l, and water salinity (total dissolved solids (TDS)) of 33.7% (337 g/l), which reaches, at lower
depths, 34.8% (348 g/l) (Salem 2009). This makes the Dead Sea the saltiest and heaviest water body on Earth and,
therefore, it is known as the 'Hyper-Saline Lake' (HSL) The Dead Sea's brine (hyper-saline water) has a uniquely
ionic composition, consisting of magnesium (1.98 mol/l), sodium (1.54 mol/l), calcium (0.47 mol/l), and potassium
(0.21 mol/l), whereas the main anions are chloride (6.5 mol/l) and bromide (0.08 mol/l). Additionally, the Dead
Sea's brine has low water activity (<0.699) (Perl et al. 2017), and its pH is approximately 6 (Oren 2010).
Until 2009 the water level (surface elevation) of the Dead Sea was 421–422 m below MSL (Abu Ghazleh et al.
2009; Salem 2009; Abu Ghazleh et al. 2010; Abu Ghazleh et al. 2011), and currently it is around 430 m (1,410 ft)
below MSL (Yechieli et al. 2016; Pletcher 2018; Witman 2018). These characteristics have made the Dead Sea the
lowest point and deepest Hyper-Saline Lake on Earth.
The Dead Sea region is known by its unique climate, as it has more than 330 sunny days, and its average
temperature is around 40 °C in summer and around 15 °C in winter, and the mean relative humidity ranges between
34% and 50% over the 12 months of the year. The annual rainfall over the Dead Sea region is around 90 mm,
meanwhile the annual evaporation rate is 1,500 mm, with an actual evaporation rate of 1,300–1,600 mm/yr,
depending on the salinity and temperature variations at the surface of the Dead Sea, which both are affected by the
annual volume of freshwater inflow into the Dead Sea. These evaporation rates indicate that an average deficit of
about 1,400 mm of the Dead Sea water occurs every year (Salem 2009). The high rates of evaporation result in the
actively precipitation of the halite and gypsum minerals, as a response to the negative water balance of the HSL,
because the evaporation rates are much greater than the inflows rates (Steinhorn, 1983; Lensky et al. 2016; Sirota et
al. 2016), which results in greater salinity (Reznik et al. 2009). The crystallization and precipitation of the halite and
gypsum minerals and salt layers in the Dead Sea and its surrounding area have been comprehensively investigated
by many researchers over the last few decades (e.g. Ganor and Katz 1989; Stiller et al. 1997; Nehorai et al. 2013;
Charrach 2018a), as well as in different salty environments in various locations around the world (e.g. Filippi et al.
2011; Bąbel and Schreiber 2016; Forti 2017).
The Dead Sea area is still home of rare species. In the mountains, oases, marshes, and temporary rivulets
surrounding the Dead Sea, there are many animals, including leopards, ibex, antelope species steenbok, and the
griffon vulture, as well as hundreds of bird species. The Jordan Rift Valley (JRV) and the Dead Sea Basin (DSB) are
among the most important migration routes for the black and white stork and many other bird species on their
migration route from the breeding areas in Eastern Europe and the Middle East to Africa (GNF 2018).



The Dead Sea is an extremely stressful hyper-saline environment and a unique model for tracking evolutionary
dynamics of biodiversity under increasing salinity. The stress of the high salinity of the Dead Sea eliminates most
life forms except one alga, several species of Archaea, bacteria, and filamentous fungi (Perl et al. 2017). Species'
diversity has steadily decreased – a phenomenon that is highly and significantly correlated with declining the water
level of the Dead Sea, as well as with increasing the water's density and salinity, which is currently 348 g/l (as
mentioned earlier). Two Dead Sea's surviving species – Aspergillus amstelodami Thom et Church and Aspergillus
ruber Thom et Church – increase in frequency in the Dead Sea, down to a depth of 291 m below MSL, due to their
evolved adaptations to tolerate hyper-salinity (Perl et al. 2017).
The Dead Sea also houses some unique bacteria, though its water is 'hyper-saline' (or brine). During December
1941, a number of samples of sediments were taken at depths of 70–330 m below MSL, which indicate the presence
of some kinds of bacteria in the Dead Sea (Volcani 1943). Micro-organisms that inhabit hyper-saline lakes may be
halo-tolerant or halo-philic. The ability to tolerate high-salt concentrations without compromising growth is
characteristic of halo-tolerant micro-organisms. Micro-organisms that have adaptations, which require salt as a
growth factor, are referred to as halo-philic. Both halo-philic and halo-tolerant micro-organisms perform one of two
different mechanisms to ensure that they can persist in the high-salt concentrations of hyper-saline environments.
Ionescu et al. (2012) discovered several underwater fresh to brackish water springs in the Dead Sea, harboring dense
microbial communities. This can be seen on the YouTube video (2.46 minutes' long) provided on this Link
<https://www.youtube.com/watch?v=aoXddPg4lFw>. This short video shows the *'First Scientific Diving Expedition*
*in the Dead Sea: Springs of Life in the Dead Sea,'* which led to the discovery of a complex community of living
microbes found in freshwater springs on the bottom of the deepest Hyper-Saline Lake (HSL) on Earth. To locate and
study these springs, it was quite a task for the scientific diving team, as the high-salt concentration makes the diving
dangerous and difficult. The divers located the springs and took samples of water and sediments, in which they
detected novel micro-organisms (MPIMM 2011).
**3.3 Dead Sea as a Health Resort**
The Dead Sea is the only place on Earth where one can sunbathe for long periods of time with little or no sunburn,
because harmful ultra-violet rays are filtered through three natural layers, which are: 1) An extra atmospheric layer;
2) An evaporation layer that exists above the Dead Sea; and 3) A rather thick Ozone Layer (OL). However, some
recent studies indicate that there is depletion in the OL over the Dead Sea, which is due to chemical effects.
Measurements of ozone ($O_3$) and bromide (or bromine oxide – BrO) concentrations over the Dead Sea indicate that
Ozone Depletion Events (ODEs), widely known to happen in polar regions, are also occurring over the Dead Sea,
due to the very high bromine content of the Dead Sea's hyper-saline water (Smoydzin and von Glasow 2009).
Bromide could play a significant role in the interaction dynamics on the surface of crystallized sea salts (Kolev et al.
2013), which may result in interactions with the OL. The bromide enrichment of the salt surfaces can play an
important role in some global atmospheric processes, like depletion of the atmospheric OL (Ghosal et al. 2000).



### 3.4 Paleo-Climate of the Dead Sea Basin

Nearly 305 m (1,000 ft) below the bed of the Dead Sea, scientists have found recently evidence that during past warm periods, the Mideast has suffered drought on scales never recorded by humans—a possible warning for current times (Kiro et al. 2017; LDEO 2017). Thick layers of crystalline salt, termed by Talbot et al. (1996) as 'salt reefs,' show that rainfall plummeted to as little as a fifth of modern levels during the Later Quaternary – some 120,000 years ago (Pleistocene Epoch on the Geological Time Scale), and again about 10,000 years ago (Holocene Epoch on the Geological Time Scale). Today, the region is drying again due to global warming, resulting from climate change, which is even getting worse. The Holocene Epoch (also known as the 'Anthropocene Epoch') is the current period of geologic time, whereas its primary characteristic is the global changes caused by human activity (or anthropogenically). The Holocene Epoch began 12,000 to 11,500 years ago at the close of the Paleolithic Ice Age and continues through today (Bagley 2013).

Charrach (2018b) studied the geological history and paleo-climate of the Dead Sea's region based on drill-holes' data, and found that a composite stratigraphic column for the Holocene Epoch, of multiple lime carbonate and halite sedimentation, has been constructed for the Southern Dead Sea Basin (SDSB). During that period of time, the SDSB has subsided at a rate of 8.5–11 m/ka (meter per a thousand years), while the subsidence of the Northern Dead Sea Basin (NDSB) may have reached 25–30 m/ka. The Holocene has been divided into 13 major climatic intervals, starting with a very arid climate from ≈ 11,700 to ≈ 8,800 BP (Before Present), where halite precipitated in both basins (NDSB and SDSB), with the Dead Sea possibly reaching a level of ≈ 419 m below MSL. After ≈ 8,800 BP, there was a very intense pluviatile period, with the formation of alluvial fans opposite wadi channels, reaching up to 45 m in thickness (Charrach 2018b).

### 3.5 Decline of Water Level in the Dead Sea

The water level in the Dead Sea has been declining since the 1950s at alarming rates of approximately one meter per year, on average (Yizhaq et al. 2017) (Figure 3 and Figure 4). The main reason for this rapid decline is the decreasing inflow of freshwater into the Dead Sea, which has been reduced from around 1,250 million cubic meter/year (MCM/yr) in the 1950s to approximately 260 MCM/yr in 2010 (Salem 2009), representing less than 21% of the original flow. In addition, the surface area of the Dead Sea has been dramatically shrinking as a result of the water level decline (Table 1).

**Table 1: Observed and projected variations of the mean sea level (MSL) and surface area of the Dead Sea over a period of 90 years (1960–2050) (Salem 2009).**

| Year | MSL (m) | Surface Area (km²) |
|------|---------|--------------------|
| 1960 | -390 | 1,020 |





| 2005 | -420 | 635 |
| 2050 | -500 | 520 |

Table (1) demonstrates that within 90 years only (1960–2050), the water level of the Dead Sea has and will be
declined by no less than 110 m (this is equivalent to more than 1.2 m/yr, on average). In absolute terms, the Dead
Sea's level has declined by 37 m as of 2017, and is forecast to drop a further 25–70 m by year 2100 (Yechieli et al.
1998; Asmar and Ergenzinger 2002; Gertman and Hecht 2002; Watson et al. 2018). However, simulations based on
ranges of water withdrawal's scenarios suggest that the Dead Sea will not "die"; rather, a new equilibrium is likely
to be reached in about 400 years after a water-level decrease of 100 to 150 m (Yechieli et al. 1998). Also, as shown
in Table (1), the surface area of the Dead Sea, for the same period (1960–2050), has and will be shrunk by about 500
km$^2$ (this is, on average, 5.5 km$^2$/yr). Abu Ghazleh et al. (2009) gave an average of 4 km$^2$/yr, as they did not
consider the projected variations in the Dead Sea's surface area until 2050.



**Figure 3: Satellite images revealing the Dead Sea's retreat (Left: 15 September 1972; Middle: 27 August 1989; Right: 11 October 2011), as well as the growth of mineral-extraction evaporation ponds in the Southern Basin of the Dead Sea (after NASA Earth Observatory 2012).**

Of the approximately water annual inflow of 1.3 billion cubic meters per year (BCM/yr), which used to naturally flow in the Jordan River, ending in the Dead Sea, more than 96% is diverted for agricultural and domestic uses by the neighboring countries (mainly Israel and Jordan, and Syria to a lesser extent), leaving only a very small amount of water to reach the Dead Sea. In addition to the diversion of the Jordan River's waters by Israel and Jordan, solar evaporation carried out by the Israeli and Jordanian mineral extraction companies on both shores (on the east and



west) of the Dead Sea along its Southern Dead Sea Basin have contributed to the drastic decline of the water level in
the Dead Sea and the decrease in its surface area and, thus, shrinking its extension (Figure 3 and Figure 4).

## The Shrinkage of the Dead Sea in the No-Project Scenario

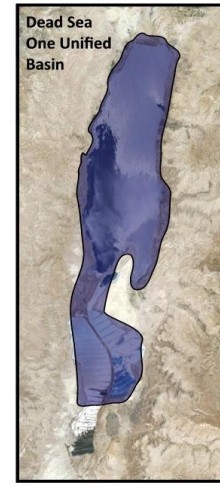 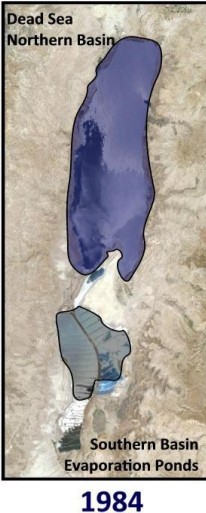 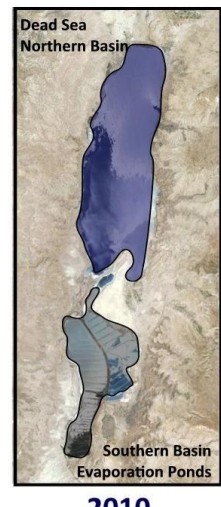 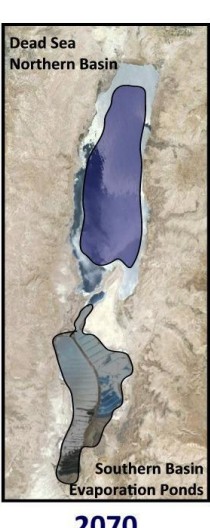

Sources: The World Bank, 2012 & The Samuel Neaman Institute, 2007

**Figure 4: Shrinkage of the surface area of the Dead Sea as a result of the decline in its water level since 1931 through the**
**years 1984 and 2010, and as projected for the year 2070 (after Willner et al. 2013).**
A great portion of the fresh water that used to recharge the Dead Sea through the Jordan River has been replaced, for
many years now, by domestic and industrial sewage that continues to flow into the Dead Sea, causing a great
damage to the Dead Sea itself and the Jordan River, as well as to the unique ecosystem of the Dead Sea Basin and
the surrounding environment. As the water level of the Dead Sea is steadily declining at alarming rate (≈1 m/yr), its
surface area is also considerably shrinking (Table 1; Figure 4). Based on satellite's imagery for a period of 41 years
(1972–2013), El-Hallaq and Habboub (2014) estimated that the Dead Sea's water area shrank on average at a rate of
≈ 2.9 km$^2$/yr.
**4. Results and Discussion**
The Dead Sea represents a unique ecosystem, with geological, chemical, biological, physical, environmental, and
ecological characteristics that are found nowhere else on Earth. The combination of these elements of uniqueness
and the special characterizations of the Dead Sea, with respect to, among others, climate, hyper-salinity, mineralogy,



topographical and geological settings, seismicity, hydrology, hydrogeology, hot springs, biodiversity, archeology,
etc. have turned the Dead Sea into a major health resort, with particularly beneficial effects on skin diseases. They
also have turned the Dead Sea Basin (DSB) into a 'Mecca' for research scientists, locally, regionally, and
internationally. Among the major concerns for research scientists in the DSB are tectonics, geodynamics, seismicity,
hyper-salinity, decline of its water surface, shrinkage of its size, and sinkholes, which all, by way or another, are
directly or indirectly related to each other and are impacted by each other. The following section discusses the issue
of DSB's sinkholes, in particular, from various points of view, as being the main target of this paper.
**4.1 Sinkholes**
**4.1.1 Anthropic Interferences and How to Deal with Sinkholes**
The high anthropic (anthropogenic) interferences and the high susceptibility of the Dead Sea's area (for health
reasons and tourism) ask for an interdisciplinary research to investigate the man-made problems that have resulted in
the occurrence of thousands of sinkholes in the region of the Dead Sea Basin (DSB). Additionally, the sinkholes'
phenomenon in the Dead Sea region is investigated in this study, based on the fact that the DSB and its tectonic and
geodynamic settings – primarily the Dead Sea Fault System (DSFS) – are considered globally an open natural
laboratory for tectonics, seismology, geology, hydro-sciences, and various disciplines of engineering. Furthermore,
the sinkholes are investigated in this study because of safety reasons, based on the fact that these sinkholes can
open-up suddenly and without warning, causing fear, panic, and anxiety, as well as death to people, and damage to
property. Therefore, those who plan roads, constructions, and infrastructure should carefully choose where to put
things, considering the sinkholes existing and others that may develop and occur anytime in the region. Because the
development and occurrence of sinkholes not only impact local infrastructures and facilities, as well as well-being of
humans, but also affect the area's hazards and risks, early monitoring of sinkhole is needed, in order to protect loss
of life and severe economic damages that may result from sinkholes (e.g. Argentieri 2015; Intrieri et al. 2015; Billi
et al. 2016; Lee et al. 2016). In addition, remarkably the sinkholes have become home for new micro- and macro-
organisms, some of which are new to the habitat of the Dead Sea Basin. The sinkholes provide a home for unusual
plant and animal communities and a special link between the Earth's surface and underground resources (Friend
2002; Keiller 2010; Adar et al. 2014).
**4.1.2 Occurrence of Sinkholes in the Dead Sea's Basin: Frequency**
The sinkholes in the Dead Sea Basin (DSB) were first noticed in the 1970s (Abelson et al. 2006), and since then they
have rapidly increased in number and size. Currently they count on both western and eastern shores of the Dead Sea
in thousands, varying in diameter, depth, size, and shape, whereas some of which are vertiginous openings tens of
meters deep (Figure 5). The largest sinkhole in the Dead Sea area has a circular shape with a diameter of 60 m and a
depth of 35 m, as well as a volume of approximately 100,000 m$^3$ (Frumkin and Raz 2001; Rybakov et al. 2001).



Though the exact number of the sinkholes in the Dead Sea area is unknown, estimates indicate that the number of
sinkholes on both sides of the Dead Sea (east and west) has probably exceeded 6,000 (Bardsley 2017). Until 2015,
the number of sinkholes on the western shore of the Dead Sea reached more than 4,000 formed since the 1970s
within a 60-km long and 1-km wide strip, and the formation rate of sinkholes in the Dead Sea's region has
accelerated in recent years to more than 400 sinkholes per year (Yechieli et al. 2016).
**Figure 5: Pictures of some examples of the sinkholes in the Dead Sea's area.**
However, the majority of the Dead Sea's sinkholes are mainly located in its southern part and on both sides of the
'Lisan Peninsula' of the Dead Sea. These sinkholes are observed mainly along the edge of a salt layer deposited
during the latest Pleistocene, when Lake Lisan receded to later become the Dead Sea (Frumkin et al. 2011). The
presence of more sinkholes in the southern part of the Dead Sea could be attributed to a man-made reason, which is
the heavy industry in the region as it has been progressively covered by solar evaporation ponds for exploitation of
the Dead Sea minerals on the western and eastern shores of the Dead Sea. It could be also attributed to a natural
cause, as tectonic deformations in this region are more apparent, because of uplift in the salt-diapir. These two
reasons (anthropogenic and natural) together might help in generating more sinkholes in the southern part of the
Dead Sea.
**4.1.3 Occurrence of Sinkholes in the Dead Sea Basin: Morphology**
In the 1960s, the Dead Sea's area was 1,200 km$^2$ (80-km long times 15-km wide). Since then the Dead Sea's area
has reduced by almost half; being now around 605 km$^2$ (Salem 2009; World Bank 2009; Tahal Group 2010; EBE
2012; Allan et al. 2014; CEB 2014; Wikipedia 2019). This has resulted in major changes in the hydrogeological
setting and in the landscape and morphology of the Dead Sea, which are direct reasons of the formation of the





sinkholes that have occurring since then at an alarming rate in the Dead Sea region.  The ground collapses and
subsidence in the Dead Sea region, characterized by underground drainage systems, have resulted in sinkholes, as
well as caves and cavities. The geomorphologic features of the sinkholes in the Dead Sea region, caused by ground
collapses and subsidence, represent an exceptional case, since they are primarily and indirectly anthropogenically
originated (Frumkin and Raz 2001).
**4.1.4 Occurrence of Sinkholes in the Dead Sea Basin: Hydrology and Hydrogeology**
As the Dead Sea's water level declines and its surface area shrinks (Figure 3 and Figure 4), the temperature of the
Dead Sea's surface water has also changed. Using observations from 10 Moderate Resolution Imaging Spectro-
radiometer (MODIS), positive trends were detected in both day-time (0.06 °C/yr) and night-time (0.04 °C/yr) Dead
Sea's surface temperature over the period of 2000–2016 (Kishcha et al. 2018). This implies greater rates of
evaporation and, thus, greater declines in the water level of the Dead Sea and more shrinkage in its surface area,
resulting in developing more sinkholes in the region.
As the brine (hyper-saline water) of the Dead Sea recedes, fresh groundwater moves up and dissolves layers of salt,
creating large underground cavities, above which sinkholes form. The decrease in the volume of brine in the Dead
Sea has pushed the fresh-water to move from the neighboring groundwater aquifer systems on the eastern and
western sides of the Dead Sea, replacing the brine. This has resulted in dissolving the salty deposits or layers, which
has led to the subsidence and collapse of the rock formations and, thus, to the formation of sinkholes with tens of
meters in diameter and in depth, along the eastern and western shorelines of the Dead Sea. These sinkholes cluster
mostly in specific sites up to 1,000-m long and 200-m wide, which align parallel to the general direction of the Dead
Sea Fault System (DSFS), associated with the Dead Sea–Jordan Valley Rift (JVR). As seen in Figure 6, the dramatic
decline of the water level in the Dead Sea for the period of approximately 40 years (≈ 1978–2018) is strongly
associated with a dramatic increase in the number of the sinkholes occurred in the region. This represents strong
evidence between the two phenomena: the decline of the Dead Sea's water level and the frequent increase of the
number and size of the sinkholes in the region.


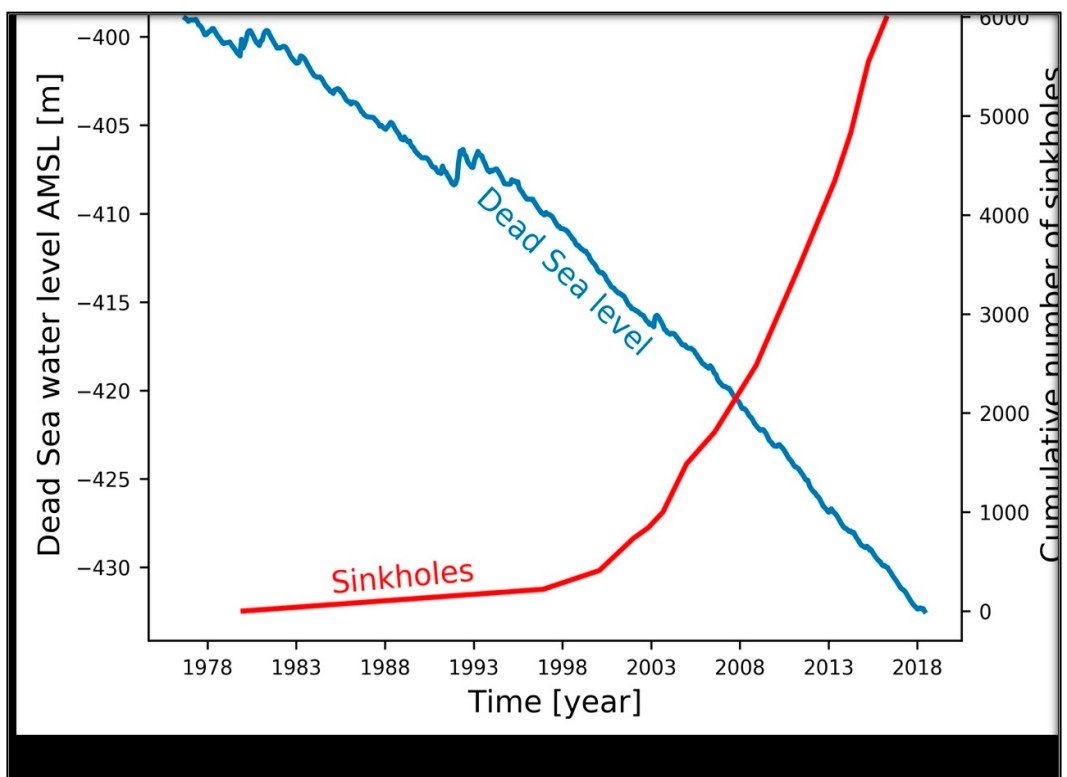

**Figure 6: Cumulative number of sinkholes (Red Line) and the decline of the Dead Sea's water level (Blue Line) for a**
**period of approximately 40 years (≈ 1978–2018) (after Nof et al. 2019).**

Over the last decade or so, the World Bank led an initiative to carry out a mega project to bring water, through a
conduit with a total length of approximately 200 km along the Wadi Araba (Arava Valley), from the Red Sea to the
Dead Sea, for the purpose of elevating the Dead Sea's water level that has been caused as a result of man-made acts
(Salem 2009; Tahal Group 2010; EBE 2012; Allan et al. 2014; CEB 2014; World Bank 2014). Nevertheless, such a
project, known as the 'Red Sea–Dead Sea Conveyance (RSDSC) project,' with the cost of billions of USD, will
never be able to provide enough water to stem the continued decline in the level of the Dead Sea. Whatever provided
of saline water from the Red Sea to the Dead Sea will be a very tiny amount of what the Dead Sea really needs. And
on the short-run and long-run, the project will create more problems than solving the already existing ones (Salem
2009). Furthermore, the addition of Red Sea's water would exacerbate the sinkhole problem. As the more diluted
water dissolves stores of salt on the shoreline, it could get absorbed into aquifers and streams, accelerating the
collapses, resulting in creation of new sinkholes and in widening the existing ones.
**4.1.5 Occurrence of Sinkholes in the Dead Sea Basin: Seismology (Natural and Induced Earthquakes)**



The features of heavy tectonics (including faults – mainly the Dead Sea Fault System (DSFS) –, ruptures, fractures,
flexures, anticlines, synclines, etc.) and seismicity (seismic activity resulting in earthquakes with small and large
magnitudes) that affect the Dead Sea's region could also be triggers behind the creation and evolution of the
sinkholes in the region (Figure 7). To investigate this and approve it scientifically, some advanced measurements are
needed to be carried on in the region. In some other parts of the world, as discussed above, it was shown that the
occurrence of sinkholes in seismically active regions is accompanied by earthquakes. As also indicated above,
infrastructures and other kinds of projects may trigger sinkholes. Salem (2009) concluded that the Red Sea–Dead
Sea Conveyance (RSDSC) project, though it has some advantages, has also a lot of disadvantages. Accordingly,
large projects that may be undertaken in the region, such as the DSRSC project, could be a reason that triggers
earthquakes and, thus, causing the creation of new sinkholes and worsening the situation of the already existing
ones. The sinkholes tend to develop along lineaments (Abelson et al. 2003), which can be traced up to ~2 km. The
orientations of the sinkhole lineaments are strikingly similar to the orientation of the faults forming the Dead Sea
Rift. These two observations imply that the sinkholes' formation is related to tectonic faults buried in the Rift's
sediments (Abelson et al. 2003). The observed linkage between tectonic faults and sinkholes implies genetic
relationships, where beside the presence of salt layer, the formation of sinkholes is strongly affected by the presence
of a prominent tectonic fault, which is the DSFS (Tahal Group 2010).

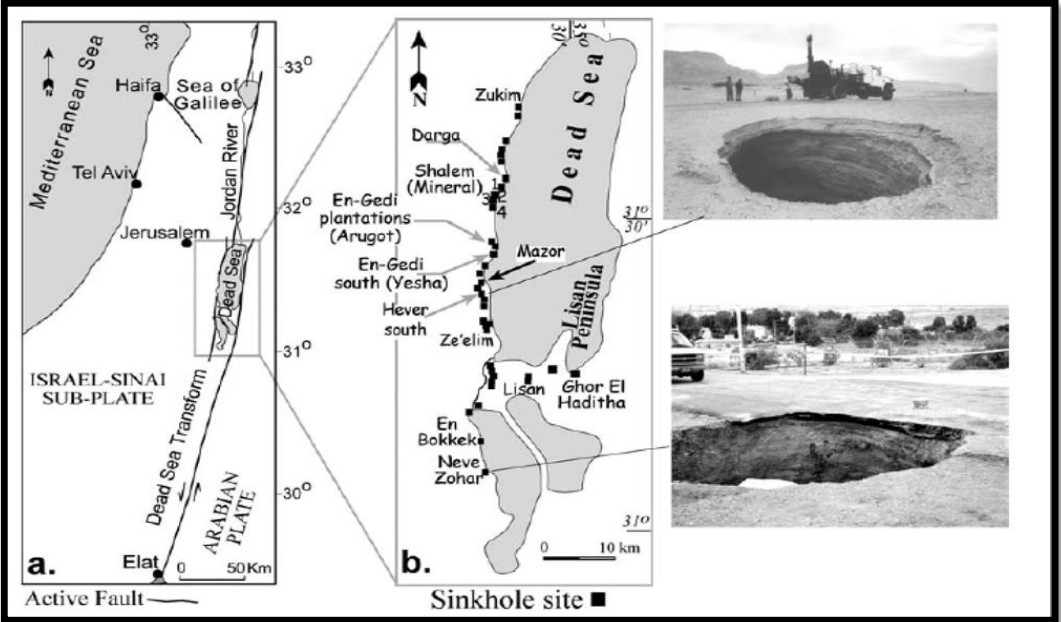

**Figure 7: Left: Location map of the Dead Sea Basin (DSB) and the Dead Sea Transform Fault (DSTF) system; Middle:**
**Distribution of sinkhole sites along the Dead Sea western shore; Right: Pictures of some sinkholes in the area, which may**
**exceed 15 m in depth and 25 m in diameter (after Abelson et al., 2003; Sherman and Rybacov 2009; Salem 2009).**



**4.1.6 Occurrence of Sinkholes in the Dead Sea Basin: Seismic Velocity Variations**

In addition to the water level's decline in the Dead Sea that definitely causes sinkholes in the Dead Sea region, as well as the earthquakes that take place in the region and that potentially cause sinkholes in the region, seismic observations, in terms of seismic velocity variations, also indicate acceleration of the development of sinkholes in the region. Geophysical studies (e.g. Ezersky et al. 2013) showed that sinkholes in the Dead Sea Basin are associated with a particular layer of halite (salt mineral), deposited 10,000 years ago (Yechieli et al. 2006). Based on in-situ seismic measurements, the halite layer exhibits a broad range of compressional wave (P-wave) velocity ($V_p$), between 2,800 and 3,600 m/s, reflecting the texture of the salt layer, which varies between solid and crumbly (Yechieli et al. 2006). On the other hand, the shear wave (S-wave) velocity ($V_s$) in the Dead Sea's halite layer ranges from 950 to 1,600 m/s (Ezersky and Frumkin 2013). These ranges of the P-wave and S-wave velocities result in values of velocity ratio ($V_p/V_s$) ranging from 2.25 to 2.95. It is worth mentioning that small differences from those obtained by Yechieli et al. (2006) for the ranges of the $V_p$ and $V_s$ values are noticed by Ezersky and Goretsky (2014).

For the Dead Sea's halite layer, the relatively low values of $V_p$ (with the range of 2,800–3,600 m/s) and the relatively low values of $V_s$ (with the range of 950–1,600 m/s) result in a relatively high values of $V_p/V_s$ (with the range of 2.25–2.95). Meanwhile, other salt layers in other regions of the world (e.g. Bourbié et al. 1987; Zong et al. 2015), and salt minerals (halite and sylvite) that were tested in laboratory under high pressure (Sun 1994), exhibit $V_p$ values in the range of 4,500–5,500 m/s and $V_s$ values in the range of 2,500–3,100 m/s. These $V_p$ and $V_s$ values result in relatively low values of $V_p/V_s$, ranging from 1.77 to 1.80. A comparison between these results indicates that the geological subsurface layers of the Dead Sea Basin are characterized by lower degrees of compaction and solidification, as well as by relatively greater values of porosity ($\varphi$) and permeability (or hydraulic conductivity) ($k$), and by lower values of tortuosity ($\tau$) (e.g. Salem and Chilingarian 2000a; Salem and Chilingarian 2000b; Lee 2003; Barrande et al. 2007; Pisani 2011; Ghanbarian et al. 2013; Ezersky and Goretsky (2014); Abderrahmene et al. 2017). The variations in these fluid-flow physical parameters ($\varphi$, $k$, $\tau$) are greatly affected by the dissolution of the salt layers in the Dead Sea Basin, which result in relatively lower values of the seismic velocities ($V_p$ and $V_s$) and higher values of the seismic velocity ratio ($V_p/V_s$). According to Ezersky and Goretsky (2014), geophysical data testify that in-situ porosity calculated, using relationships obtained during their study, indicate zones of heightened voidness (where $\varphi > 25\%$). They also noticed that permeability (hydraulic conductivity) ($k$) in the same zones are high and, thus, they expected that in larger salt volumes these parameters can yet increase.

In seismic imaging, the salt body is often assumed to be isotropic and homogeneous with constant velocities (Fuyong et al. 2016). The large variations in the P-wave velocity (2,800–3,600 m/s) and the S-wave velocity (950–1,600 m/s), resulting in large values of the velocity ratio (2.25–2.95) indicate anisotropy and heterogeneity, characterizing the salt layer in the Dead Sea Basin. These indications of anisotropy and heterogeneity of the Dead



Sea's salt layers agree well with the results obtained by Hatzor and Heyman (1997) from laboratory tests on salt
(halite) samples taken from the salt deposits in the Dead Sea region. Anisotropy, associated with heterogeneity, as in
the case of the Dead Sea's halite (salt) layer, means variations in physical properties in the three different directions
($X,Y,Z$) of the salt layer, such as variations in hydraulic, electric, and heat flow, as well as in propagation of seismic
wave (acoustic signal) velocity (Salem 1994).
**4.1.7 Occurrence of Sinkholes in the Dead Sea Basin: Geochemistry-Fluid Dynamics**
Shalev et al. (2006) showed, through finite-element modeling, that dissolution of the salt layer(s) in the Dead Sea
Basin is a plausible mechanism to explain the rapid creation of subsurface holes that collapse, forming sinkholes.
The positive interaction among the rate of flow, the rate of chemical reaction, and the change in permeability
(hydraulic conductivity) accelerates the dissolution processes and might result in 'reactive infiltration instability,'
which is manifested in interconnected cavities, into which fluid is channeled, as a result of salt dissolution. The
frequent occurrence of sinkholes, the spacing between them, and the high rate of their development and formation in
the Dead Sea Basin are controlled by several factors. These include: properties of structural elements (lineaments),
such as faults, ruptures, fissures, flexures, channels, voids, etc.); flux and freshness of incoming groundwater; rate of
dissolution; effective specific surface area of particles (e.g. Salem and Chilingarian 1999); permeability (hydraulic
conductivity) of the salt and clay layers; the $\varphi$–$k$–$\tau$ relations or their dependence on each other; dispersivity; and
thickness of the layers. The salt's dissolution in the underlying layers results in higher values of porosity, higher
values of permeability, and lower values of tortuosity (as indicated above), which all lead to easy flow of fresh water
into the salt layers. Thereby, this will lead to increasing the rates of both solute transport and the chemical reactions,
resulting in fluid channeling and, thus, cavitations, which leads to easy collapse, resulting in huge sinkholes. A large
number of sinkholes' sites occurred where both the edge of the halite (salt) layer and underground discontinuities
(faults or fractures, acting as preferential channel-ways) are simultaneously present (Ezersky and Frumkin 2013).
However, these three fluid-flow parameters ($\varphi$–$k$–$\tau$) need to be measured, in order to further understand the impacts
of fluid dynamics (fresh-water and saline-water) on the formation of the sinkholes in the Dead Sea Basin.
**5 Conclusions and Recommendations**
Sinkholes—large, open holes that result from the collapsing of Earth's surface—represent serious environmental
and geotechnical problems in the Dead Sea Basin (DSB). The sinkholes began to appear in the DSB about 20 years
after the Dead Sea's water level started to decline. The 20-year delay in the occurrence of the sinkholes after the
recession of the Dead Sea began means it takes 20 years for the freshwater to flush all the way to the salt layers and
then to dissolve the salt, creating cavities within the salt layers, up to the shallower clay and gravel layers. This
indicates that there is a strong correlation between both phenomena: the decline of the Dead Sea's water level, on
the one hand, and the remarkable and frequent occurrence of the sinkholes, on the other. This means that the greater



the decline of the Dead Sea's water level, the greater the number and the larger the size of the sinkholes that have
been already developed and others that will develop anytime soon in the Dead Sea Basin.
In addition, the tectonics and seismic activity that affect the Dead Sea Basin, besides the decline of the water level in
the Dead Sea and the shrinkage of its surface area, might trigger the occurrence of plenty of sinkholes in the region.
The tectonic features existing in the region (faults, fissures, ruptures, fractures, flexures, etc.) serve as conduits,
channeling freshwater from the deeper aquifer systems to the shallower ones, dissolving the salt layers and, thus,
promoting the development of the sinkholes. This implies that more voids are created and, thus, higher porosities
and hydraulic conductivities are resulted, which lead to the movement of greater amounts of freshwater. Once
porosity and permeability (hydraulic conductivity) are increased by dissolution, fluids are channeled into the
dissolved sections and accelerate the process, as the salt layers become more porous and more permeable and, thus,
less tortuous, leading to easy flow of fresh water into the layers. In other words, the heterogeneity created by
dissolution magnifies this instability and leads to more and larger sinkholes in the Dead Sea Basin.
The factors discussed in this paper (including decline of water level, shrinkage of surface area, decrease of water
volume, tectonic features, seismicity (or earthquake activity)), affecting and will affect the Dead Sea Basin, have led
and will further lead to the occurrence of sinkholes in the Dead Sea's region. Physically, chemically, and
hydrogeologically, the impacts of these phenomena are reflected on the variations in acoustic (seismic) wave
propagation and fluid-flow parameters, such as seismic wave velocity, velocity ratio, porosity, permeability,
tortuosity, and others. Analyses of available data of the compressional and shear wave velocities, as well as the
fluid-flow parameters for the Dead Sea Basin indicate that the relatively low values of in-situ velocities of both
kinds of seismic waves result in greater values of the compressional to shear velocity ratio. These observations
indicate heterogeneity and anisotropy, as well as less compaction and solidification of the layers of the DSB, which
are all encouraging factors for the sinkholes' development and formation in the Dead Sea's region.
Some of the factors mentioned above are anthropogenic (including the decline of water level of the Dead Sea and
shrinkage in its area and, thus, decrease in its water volume, and some others are natural (including tectonic features
and seismicity), which have resulted in the formation of already existing sinkholes and more of them in the near
future. However, the anthropogenic factors could be avoided, if man really pays more attention to nature and the
environment, especially when it comes to a unique feature like the Dead Sea Basin, including the Dead Sea itself,
the Jordan Rift Valley, and their surrounding region.
The Dead Sea shorelines are undergoing continuous damages and eco-hazards, mainly due to the decline of the
water level in the Dead Sea, because of anthropogenic impacts, and also because of the intensive physical
infrastructures, building of constructions (such as hotels, etc.) and potential mega projects (such as the Red Sea-
Dead Sea Conveyance project), deepening and deforming drainage systems, landslides, land subsidence, pollution of
the Dead Sea's coasts and its coastal springs, and formation of the sinkholes. Several alarming cases happened in the



last few years in the Dead Sea's region, as a result of the collapse and subsidence of the Earth's surface, resulting in
huge sinkholes. Subsequently, destination for recreation and tourism has been considerably decreased, because of
warnings related to further sinkholes and, thus, tourists are scared from further disasters and eco-hazards that may
take place in the region, anytime.
Based on the fact that the sinkholes' phenomenon has created a state of fear and panic in the region, which has
already affected the recreation and tourism businesses, measurements and monitoring actions need to be steadily and
permanently conducted in the region. In addition, an early warning system needs to be installed in the region, so that
it can provide dual services, i.e. in relation to the seismic activity that result in micro- and macro-earthquakes that
frequently hit the region, and also in relation to the sinkholes that frequently occur in the region. Field and
laboratory measurements, related to sinkholes and others forms of Earth's surface collapses and subsidence in the
Dead Sea Basin, should be done through geophysical and other kinds of techniques. These may include seismic,
electric, electromagnetic, and geo-radar, as well as Interferometric Synthetic Aperture Radar (InSAR), Differential
Interferometric Synthetic Aperture Radar (DInSAR), Geographic Information System (GIS) and Remote Sensing
(RS), Satellite Image Time Series (SITS), Moderate Resolution Imaging Spectro-radiometer (MODIS), Terrestrial
Laser Scanner (TLS), Spatial Multi-Criteria Evaluation (SMCE), and Light Detection and Ranging (LiDR), along
with detailed geomorphological, geological, seismological, and limnological studies and surveys.
**Abbreviations and Units**
BP              Before Present
BrO             Bromide Oxide
DSB             Dead Sea Basin
DSF             Dead Sea Fault
DSFS            Dead Sea Fault System
DSTF            Dead Sea Transform Fault
FDEM            Frequency Domain Electro-Magnetic
GIS             Geographic Information System
GPR             Ground-Penetrating Radar
HSL             Hyper-Saline Lake
InSAR           Interferometric Synthetic Aperture Radar
JRV             Jordan Rift Valley
LiDAR           Light Detection and Ranging
MODIS           Moderate Resolution Imaging Spectro-radiometer
$O_3$           Ozone
ODEs            Ozone Depletion Events
OL              Ozone Layer



| | | |
|---|---|---|
| 1 | P-wave | Compressional Wave (Seismic, Acoustic) |
| 2 | RS | Remote Sensing |
| 3 | RSDSC | Red Sea–Dead Sea Conveyance (Project) |
| 4 | SITS | Satellite Image Time Series |
| 5 | SMCE | Spatial Multi-Criteria Evaluation |
| 6 | S-wave | Shear Wave (Seismic, Acoustic) |
| 7 | TDS | Total Dissolved Solids |
| 8 | TLS | Terrestrial Laser Scanner |
| 9 | USD | United States' Dollar |
| 10 | $V_p$ | Compressional Wave Velocity |
| 11 | $V_s$ | Shear Wave Velocity |
| 12 | $V_p/V_s$ | Ratio of Compressional Wave Velocity to Shear Wave Velocity |
| 13 | $k$ | Permeability (Hydraulic Conductivity) |
| 14 | $\varphi$ | Porosity |
| 15 | $\tau$ | Tortuosity |
| 16 | | |
| 17 | BCM/yr | billion cubic meters per year |
| 18 | °C | degree Celsius |
| 19 | °C/yr | degree Celsius per year |
| 20 | ft | foot |
| 21 | g/l | gram per liter |
| 22 | kg/l | kilogram per liter |
| 23 | km | kilometer |
| 24 | $km^2$ | square kilometer |
| 25 | m | meter |
| 26 | $m^3$ | cubic meter |
| 27 | MCM/yr | million cubic meters per year |
| 28 | m/ka | meter per thousand years |
| 29 | m/s | meter per second |
| 30 | mm | millimeter |
| 31 | mm/yr | millimeter per year |
| 32 | mol/l | mole per liter |
| 33 | pH | acidity–basicity Indicator |
| 34 | % | percentage |
| 35 | | |
| 36 | **Declaration** | |
| 37 | | |



The Author of this paper declares the following: 1) Compliance with Ethical Standards, including: The research
presented in herein does not involve human participants and/or animals; 2) Informed consent: There is no potential
of conflict of interest of any kind (financial or otherwise); and 3) Funding: The research presented in this paper did
not receive any funding from any individuals or organizations.
**Acknowledgements**
The Author expresses his sincere thanks to his friends and colleagues, who critically reviewed this paper and
provided very constructive comments.

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
