# Peer review of "Sinkholes of the Dead Sea Basin: A Result of Anthropogenic Disturbance to Nature and Sign for More and Greater Hazards"

_Solid Earth, 2019_

## Referee Comment (RC1) · Anonymous Referee #1 · 12 Jun 2019

Regretfully, I recommend rejection of submission se-2019-97. This manuscript does not present any new findings nor novel perspectives.

The abstract is not a proper summary although has an appropriate length. It reads like a short introduction and it does not help the reader to find out the paper's purpose. I recommend following a classic structure for the abstract: short background, methodology, detailed results, and conclusions.

In the geological introduction, the paragraphs related to the definition of a sinkhole are ambiguous. For instance, this manuscript reads '(…) caused by some forms of collapse of the surface layers because of different reasons (Goudie, 2006)' and I believe it

would be useful to follow the widely accepted definitions and classifications proposed by sinkhole experts.

According to the author the paper's main target is 'to describe this hazardous phenomenon (sinkholes of the Dead Sea) and its several social and economic implications' but I think the sections '3.2 Unique Characterizations and Biodiversity of Dead Sea Basin' and '3.3 Dead Sea as a health Resort' are not relevant to that topic.

The conclusions are not novel, and the author recommends the application of several techniques without any justification, despite most of them are already being used.

The figures are too focused on the geological and geographical context of the Dead Sea Basin and the one showing examples of the sinkholes in the Dead Sea's area (Figure 5) lacks a detailed description, location and scales to fully grasp the sinkholes illustrated.

---

## Referee Comment (RC2) · Anonymous Referee #2 · 1 Jul 2019

This is a manuscript about the sinkholes of the Dead Sea Basing derived from Anthropogenic disturbance. After reading through the manuscript, serious issues have been detected that advise against its publication. The first (and major one) is that it lacks a logical structure (a bit awkward), i.e. "study purpose and methodology" as a single (and poorly detailed) section; "data and observations" including "Geological-Geophysical Setting" (better Study area / regional setting?). Moreover, it lacks a "Results" section, and consequently, it is impossible to differentiate the results obtained by the authors from those cited in the literature. Apart from the structure, figures are nor very communicative and informative by themselves and need considerably more work (maps without any of the basic cartographical elements, i.e. scale, key, etc.; need of

homogenize the overall graphic layout style, etc.). In addition, the manuscript should have been focused in a more scientific way. For the above mentioned reasons, the paper should be rejected in its current form.